# Estimating the real burden of disease under a pandemic situation: The SARS-CoV2 case

**Amanda Fernández-Fontelo**[1]*, **David Moriña**[2,3,4], **Alejandra Cabaña**[2], **Argimiro Arratia**[5], **Pere Puig**[2]

1 Chair of Statistics, School of Business and Economics, Humboldt-Universität zu Berlin, Berlin, Germany, 2 Departament de Matemàtiques, Barcelona Graduate School of Mathematics (BGSMath), Universitat Autònoma de Barcelona, Barcelona, Spain, 3 Department of Econometrics, Statistics and Applied Economics, Riskcenter-IREA, Universitat de Barcelona, Barcelona, Spain, 4 Centre de Recerca Matemàtica (CRM), Barcelona, Spain, 5 Department of Computer Science, Universitat Politècnica de Catalunya, Barcelona, Spain

* fernanda@hu-berlin.de

**Data Availability Statement:** Data are open and available at https://cnecovid.isciii.es/covid19, https://afondo.farodevigo.es/galicia/el-mapa-del-coronavirus-en-galicia.html and

## Abstract

The present paper introduces a new model used to study and analyse the severe acute respiratory syndrome coronavirus 2 (SARS-CoV2) epidemic-reported-data from Spain. This is a Hidden Markov Model whose hidden layer is a regeneration process with Poisson immigration, Po-INAR(1), together with a mechanism that allows the estimation of the under-reporting in non-stationary count time series. A novelty of the model is that the expectation of the unobserved process's innovations is a time-dependent function defined in such a way that information about the spread of an epidemic, as modelled through a Susceptible-Infectious-Removed dynamical system, is incorporated into the model. In addition, the parameter controlling the intensity of the under-reporting is also made to vary with time to adjust to possible seasonality or trend in the data. Maximum likelihood methods are used to estimate the parameters of the model.

## Introduction

A major difficulty in the fight against the pandemic caused by the severe acute respiratory syndrome coronavirus 2 (SARS-CoV2) is the large number of people who become infected and experience a mild form of the disease but can pass it on to others [1, 2]. The lack of tests to carry out large-scale diagnoses and the different protocols regarding testing policies add an extra source of uncertainty about the true number of infected individuals. This causes the number of cases reported by the authorities that serve as a basis for public health policies to severely underestimate the actual number of cases in the population [3].

The problem of under-reporting affects data quality and therefore contributes to misrepresent results and conclusions, as reported observations do not reflect the total amount of cases of interest but only a fraction of them. Any measure related to the evolution or impact of the epidemic (e.g., lethality rates, basic and effective reproduction numbers, and others) will be distorted. This problem is not exclusive of epidemics but pervades in most areas of public

https://www.juntadeandalucia.es/
institutodeestadisticaycartografia/badea/informe/
anual?CodOper=b3_2314&idNode=42348. Codes
are available at https://github.com/underreported/
HMCmodel.

**Funding:** This work was co-funded by Instituto de
Salud Carlos III (COV20/00115), and the Spanish
Ministry of Economy and Competitiveness
(RTI2018-096072-B-I00). A.Fernández-Fontelo
acknowledges financial support from the German
Research Foundation (D.F.G.). D. Moriña
acknowledges financial support from the Spanish
Ministry of Economy and Competitiveness,
through the Mara de Maeztu Programme for Units
of Excellence in R&D (MDM-2014-0445) and
Fundacion Santander Universidades. A. Arratia
acknowledges support by grant TIN2017-89244-R
from MINECO (Ministerio de Economa, Industria y
Competitividad) and the recognition 2017SGR-856
(MACDA) from AGAUR (Generalitat de Catalunya).
This work was also processed in the frame CY
Initiative of Excellence (grant\Investissements
d'Avenir" ANR- 16-IDEX-0008), Project \EcoDep"
PSI-AAP2020-0000000013. The funders had no
role in study design, data collection and analysis,
decision to publish, or preparation of the
manuscript.

**Competing interests:** The authors have declared
that no competing interests exist.

**Abbreviations: HMM**, Hidden Markov Models;
**INAR**, integer-valued autoregressive; **MLE**,
Maximum likelihood estimates; **PCR**, polymerase
chain reaction; **SARS-CoV2**, severe acute
respiratory syndrome coronavirus 2; **SEIR**,
Susceptible-Exposed-Infectious-Recovered; **SIR**,
Susceptible-Infectious-Removed.

health, economics, and society, among others. During the past years, several authors have studied this phenomenon in different applications. Among these authors, we can highlight [4] who studied the under-reporting in worker absenteeism through Markov chain Monte Carlo analysis, [5] who considered a Bayesian approach to estimate the number of committed crimes in Málaga in 1993 and Stockholm between 1980 and 1986, [6] who described the under-reported in work-related skin diseases in Norway from 2000 to 2013, or [7] who studied under-reporting in tuberculosis in Brazil. Global percentages of under-reporting during a given period of time can be estimated, for instance, with stochastic Susceptible-Exposed-Infectious-Recovered (SEIR) models, including unobservable compartments of non-ascertained individuals. In order to estimate the parameters in such models, it is necessary to have data on individual evolution of the epidemic, that is, for each individual, the date of contagion or appearance of first symptoms, number of days in quarantine, hospital, or similar information is required, regardless of the estimation methodology. There are many examples of this situation. For instance, [8] who estimates SARS-CoV2 in Wuhan via MCMC, [9] who uses least squares estimator for SARS-CoV2 in Uruguay, or [10] who employs a simpler version of an SEIR model, called Susceptible-Infected-Recovered (SIR) model, to understand the relationship between the observed and unobserved cases of the Hong Kong seasonal influenza epidemic in New York between 1968 and 1969. Although the previous works proposed new methods to describe, identify or estimate under-reporting of data, none of them, to our knowledge, tried to model the under-reporting in integer-valued time series data.

In [11] it is proposed a simple model for integer-valued time series data that estimates the under-reporting of the human papillomavirus infections in the province of Girona from 2010 to 2014, the number of deaths attributable to a rare, aggressive tumour (pleural and peritoneal mesotheliomas) in Great Britain from 1968 to 2013, and the number of botulism cases in Canada from 1970 to 2013. The model mentioned above was extended by considering a more complex correlation structure among the time series observations in [12], where the authors studied the number of real cases of gender violence in Galicia from 2007 and 2017. The adequacy of the models in [11, 12] was assessed through simulations of different scenarios that were well recovered by the estimation procedure, and as for real data, the results coincide with the expert's opinion.

Our original motivation for this work was to study daily reported cases of SARS-CoV2 in different areas of Spain. The protocol for testing as of February 2020 only included clinically suspicious patients who recently arrived from China [13]. The protocol experienced changes in the succeeding weeks, and by May 2020, the norm became the polymerase chain reaction (PCR) or molecular tests performed to individuals with a broader collection of symptoms and contacts of confirmed patients [14]. This scenario suggests a hidden process that governs the evolution of the daily number of infected individuals, and an observed process that reflects only part of it. Moreover, the proportion of unobserved cases varies in time, due at least to the changes in testing protocols. On the other hand, it is reasonable to assume that the underlying process is non-stationary since the evolution of the epidemic of SARS-CoV2 has been observed to evolve initially drawing a mild logarithmic curve followed by an outbreak with exponential growth, which later slows down and also declines exponentially, with varying growth-decay rates which depend much on the application and effectiveness of public health prevention measures.

In light of all the above considerations, we propose here a new extension of the model in [11], which deals with the non-stationary behaviour of the hidden process and estimates the under-reporting in epidemics such as the SARS-CoV2 and potential outbreaks. The unobserved process is modeled with an INAR(1) structure, assuming that for each case counted during day $n$, a new case appears in day $n + 1$ with a fixed (yet unknown) probability $\alpha$, and to

these, a random number of other counts are added (innovations). We shall assume that these innovations are independent of the past and Poisson distributed. The mean of the innovations will be modelled as the difference of the affected individuals from day $n$ to day $n + 1$, found through the solution of a SIR (Susceptible, Infectious, and Removed) compartmental model, thus taking into account the spread of the epidemic. We reconstruct the most plausible count for each time and propose different forecasting methods. The latter allows us to estimate more precisely measures such as the lethality rate and provide more accurate predictions for applying more realistic control and prevention measures. The model is applied to the time series of the number of new daily SARS-CoV2 cases confirmed by PCR in different regions with different characteristics and climate conditions in Spain. Despite being especially useful to model and estimate the under-reporting in small areas with low counts, the application also shows that our model can be used in larger areas that can be split into smaller regions following geographic or sanitary criteria (e.g., dividing large areas into smaller sanitary areas).

## Methods

### Modelling the under-reporting of stationary time series

Our approach to the modelling of the non-reported daily counts in the SARS-CoV2 cases series is an extension of the model introduced in [11], that we briefly discuss here.

Consider that the true (unobserved) counts come from a process $X_n$, $n \in \mathbb{N}$, defined with an integer-valued autoregressive model of order 1 (INAR(1)):

$$X_n = \alpha \circ X_{n-1} + W_n, \tag{1}$$

where $0 < \alpha < 1$ is a fixed parameter, and $W_n$ are the innovations, distributed according to a discrete probability law, independent of $X_n$. The operator $\circ$ is the binomial *thinning* or *subsampling* operator defined by:

$$[\alpha \circ X_{n-1} | X_{n-1} = x_{n-1}] = \sum_{j=1}^{x_{n-1}} B_j, \tag{2}$$

where $\{B_j\}$ is a sequence of independent and identically distributed Bernoulli random variables with parameter $\alpha$, denoted as Bern($\alpha$). Note that $[\alpha \circ X_{n-1} | X_{n-1} = x_{n-1}] \sim \text{Binomial}(x_{n-1}, \alpha)$.

The model in (1) can be seen as an homogeneous Markov chain with transition probabilities given by:

$$P(X_n = i | X_{n-1} = j) = \sum_{k=0}^{\min(i,j)} \binom{j}{k} \alpha^k (1-\alpha)^{j-k} P(W_n = i - k), \tag{3}$$

where, in the case of the so-called INAR(1) process with Poisson innovations, $P(W_n = i - k) = \frac{e^{-\lambda} \lambda^{i-k}}{(i-k)!}$. The standard interpretation of an INAR(1) model is that a proportion $\alpha$ of the individuals at time $t$ "survive" and are part of the population at time $t + 1$. However, this interpretation is misleading in our context. The observations at time $t + 1$ are all new individuals; some correspond to the binomial thinning and the others to the independent innovations. It is known that for many applications for where INAR(1) models can be applied, this meaningful interpretation is not possible. However, the thinning is needed for modelling the autocorrelation of time series. For instance, this is the situation for the example of meningococcal infection analysed in [15].

More details on the INAR(1) model and several extensions can be found in [16–20] or in [21–24] where INAR models based on generalisations of the binomial thinning operators (e.g., expectation thinning operators) are defined.

Now consider a very simple mechanism that can lead to an observable and potentially under-reported process $Y_n$:

$$
Y_n = \begin{cases} X_n & \text{with probability } 1 - \omega \\ q \circ X_n & \text{with probability } \omega. \end{cases}
$$

(4)

That is, for each $n$, we observe $X_n$ with probability $1 - \omega$, and a $q$-thinning (as defined in Eq (2)) of $X_n$ with probability $\omega$, independent of the past $\{X_j: j < n\}$. Therefore, what we observe (the reported counts) is $Y_n = (1 - 1_n)X_n + 1_n \sum_{j=1}^{X_n} \xi_j$ where $1_n \sim \text{Bern}(\omega)$ and $\xi_j \sim \text{Bern}(q)$.

In the next sections, we will generalise this process to allow for non-time-homogeneous processes, by modelling the mean of the innovations in (1), as well as the under-reporting parameter $q$ in (4), as functions of time.

**Parameter estimation.** The parameters of the model can be estimated using different strategies. In [11], the authors proposed a moments-based method and a likelihood-based method. Since the first method is only appropriate when the series is stationary, we will focus on the second method of estimation based on the likelihood function.

The model described in (1) and (4) is a Hidden Markov Model (HMM) with an infinite number of states [25, 26], and hence, the maximum likelihood estimators of the parameters involve intensive numerical computations. For a given $n$, the possible values of the series $X_n$ must be equal to or greater than the observed value of $Y_n$, which implies a wide range of possible trajectories. Given the observed series, there are a countable number of potential sequences that can lead to it, and therefore the likelihood function cannot be computed directly. A way to circumvent this problem consists of using the forward algorithm [25, 26]. This recursive algorithm is linear in $n$; it is based on the forward probabilities of the Markov Chain that can be computed in terms of the transition and emission probabilities. These forward probabilities are defined by

$$
\gamma_k(y_{1:k}, x_k) = \sum_{x_{k-1}} P(Y_k = y_k | X_k = x_k) P(X_k = x_k | X_{k-1} = x_{k-1})
$$
$$
\gamma_{k-1}(y_{1:k-1}, x_{k-1}).
$$

(5)

Thus, the likelihood function of the model can be computed as
$P(Y_1 = y_1, Y_2 = y_2, \ldots, Y_n = y_n) = \sum_{X_n = y_n}^{\infty} \gamma_n(y_{1:n}, x_n)$.

In this case, the transition probabilities, $P(X_k = x_k | X_{k-1} = x_{k-1})$ are given by the Eq (3). That is, the transition probabilities are defined by the conditional probability mass function of the INAR(1) model. On the other hand, the emission probabilities are defined by:

$$
P(Y_k = y_k | X_k = x_k) = \begin{cases} 0 & y_k > x_k, \\ (1 - \omega) + \omega q^{x_k} & y_k = x_k, \\ \omega \binom{x_k}{y_k} q^{y_k} (1 - q)^{x_k - y_k} & y_k < x_k. \end{cases}
$$

(6)

Notice that, in practice, an upper threshold has to be defined in the sum that computes the likelihood function. In this application, this threshold is fixed as 1.5 times the maximum value of the series.

The reconstruction of the most likely latent sequence is a key point in the current analysis since it gives us a picture of how the unobserved process behaves. To do so, the Viterbi algorithm [27] is used, which consists of finding the sequence $X^*$ that maximises the likelihood function of $X_n$ given the observed process $Y_n$ and a known vector of parameters. That is, $X^* = \text{argmax}_X \hat{P}(X_{1:n}|Y_{1:n}) = \frac{\hat{P}(X_{1:n}|Y_{1:n})}{\hat{P}(Y_{1:n})}$. However, since the denominator $\hat{P}(Y_{1:n})$, does not depend on $X_n$, it suffices to maximise the joint probability $\hat{P}(X_{1:n}|Y_{1:n})$.

## Modelling the spread of an epidemic: The SIR model

The key interest of researchers when dealing with an epidemic such as the current SARS-CoV2 is to estimate the propagation of the disease and predict its possible end date to apply appropriate measures of control and prevention [28]. The literature offers different approaches to deal with so, as the so-called SIR and SEIR compartmental models. These models have extensively used for study influenza's epidemic evolution as [29] who use an SEIR model to evaluate vaccine policies effects on England and Wales's influenza epidemics, [30] who employs SEIR model to study seasonal influenza evolution in England by linking the prior seasonal information to the immunity in the following period in order to ensure non-independence between the successive influenza seasons, or [31] who presents a set of different research works aimed at modelling the influenza epidemics.

We shall link the expectation of the innovations in (1) to the daily number of individuals affected by the disease. For that purpose, we will study a simplified version of the SIR model. This model belongs to the class of compartmental models, and a system of ordinary differential equations governs its behaviour. Consider three classes of individuals at each time $t \in \mathbb{R}$: those who are healthy but susceptible to get the disease ($S(t)$), those who are infected and thus transmitters of the disease ($I(t)$), and those individuals who have been removed from the system and will not get infected again ($R(t)$) [32, 33]. The SIR model describes the dynamics of the spread of the virus and it is formally defined by a system of differential equations given in S1 Appendix.

The parameters of interest are $\beta$, $\gamma$, and $N$, which are the infection rate, the removal rate, and the total susceptible population, respectively. For each $t$, the following condition is fulfilled: $S(t) + I(t) + R(t) = N$. Usually, the initial conditions are set to $R(0) = 0$, $I(0) = I_0$ and $S(0) = N − I_0$. Although this model sensibly represents certain epidemics' evolution, it is hard to fit into real-world data due to the sensitiveness to slight changes in both the parameters' values and the initial conditions.

Consider now the number of affected individuals $A(t) = I(t) + R(t)$. In S1 Appendix, we show that the number of individuals affected by the disease can be fairly represented by:

$$A(t) = \frac{M^* A_0 e^{kt}}{M^* + A_0(e^{kt} − 1)}. \tag{7}$$

where $k = \beta − \gamma$ and $M^* = \frac{N(\beta − \gamma)}{\beta − \gamma/2}$. Recall that $\beta$ is the infection rate, $\gamma$ is the recovery rate and $N$ the total susceptible population.

The solution given in (7) allows to take into account the information on the spread of the epidemics in the model given by (1) and (4), by considering that the expectation of the innovations, instead of being constant, that is, $\lambda$, it will be a function of time such that $\lambda_t = \text{new}(t) = A(t) − A(t − 1)$, where $\text{new}(t)$ are the new affected cases at time $t$. It can be seen that the Eq (7)

behaves as an exponential function close to the origin, that is, $A(t) \approx A_0\, e^{kt}$ when $t \approx 0$. Therefore, the new affected cases grows exponentially at the beginning, that is, new$(t) = A(t) - A(t-1) = A_0(1 - e^{-k})e^{kt}$. In addition, the function $A(t)$ tends to $M^*$ as $t$ tends to $\infty$. The maximum value of $A(t)$, that is, $A(\infty)$ can be obtained by numerically solving the following equation (see S1 Appendix for details):

$$A(\infty) - \frac{N\gamma}{\beta}\log\left(\frac{N - A_0}{N - A(\infty)}\right) + R_0 = 0. \qquad (8)$$

Eq (8) can be especially useful in the reconstruction of the SIR process by recovering the parameters $\beta$, $\gamma$, and $N$ once the under-reporting model is estimated.

Taking into account this SIR representation of the expected value of the innovations $\lambda_t$ in the latent process model, a more realistic description of the model for the SARS-CoV2 data will be derived, which will allow estimating the characteristics of the under-reporting in such data and the spread of the epidemic jointly.

## Modelling the under-reporting of non-stationary time series including information on the spread of the disease

The model described in Eqs (1) and (4) is useful for detecting and quantifying the under-reporting at a local scale because of the likelihood computations work well with relatively small counts. It is also meant to model weakly stationary processes, i.e., with expectation, variance, and auto-covariances not varying in time.

However, many real-world time series data are non-stationary as they may be governed by trends or volatilities and may have different seasonal and cyclic patterns. For example, the series of daily new SARS-CoV2 cases analysed in the present work show intricate trend patterns. The observed series in the cases we analyse present a seasonal component due to the "weekend effect". That is, the number of cases reported during certain days of the week decreases, and thus the official records show fewer cases periodically. This behaviour repeats weekly. It is a problem attributable to the reporting process and not to the nature of the underlying phenomenon. The model in Eqs (1) and (4) has to be modified in order to take this particularities into account.

We proceed now to incorporate the information on the evolution of the SARS-CoV2 epidemic into the model in Eqs (1) and (4), and thus to fit the trend displayed in Figs 1, 3, and 4 appropriately.

Our analysis is based on the daily number of new cases of SARS-CoV2. For each time $n$, the new counts can be expressed in terms of the affected number of individuals introduced in the previous section. That is, for each $n$, the number of new individuals can be defined in terms of the number of affected individuals, as follows: new$(n) = A(n) - A(n-1)$, where $A(n)$ is defined in (7). This information can be appropriately incorporated into the model to accommodate the trend present in the data using the information on the propagation of the epidemic provided by the data themselves.

A sensible way to do this is by considering that the expectation of the innovations of the latent process $X_n$ in expression (1) varies with the number of new cases at each time $n$, and thus that the model in (1) and (4) is not stationary anymore. Specifically, the innovations of $X_n$ will have Poisson distribution with $\lambda_n = $ new$(n) = A(n) - A(n-1)$, where $A(n)$ is given by (7). Therefore, the unobserved process $X_n$ becomes:

$$X_n = \alpha \circ X_{n-1} + W_n(\lambda_n), \qquad (9)$$

where the value of parameter $\alpha$ is still fixed between $(0, 1)$, but now the parameter of the

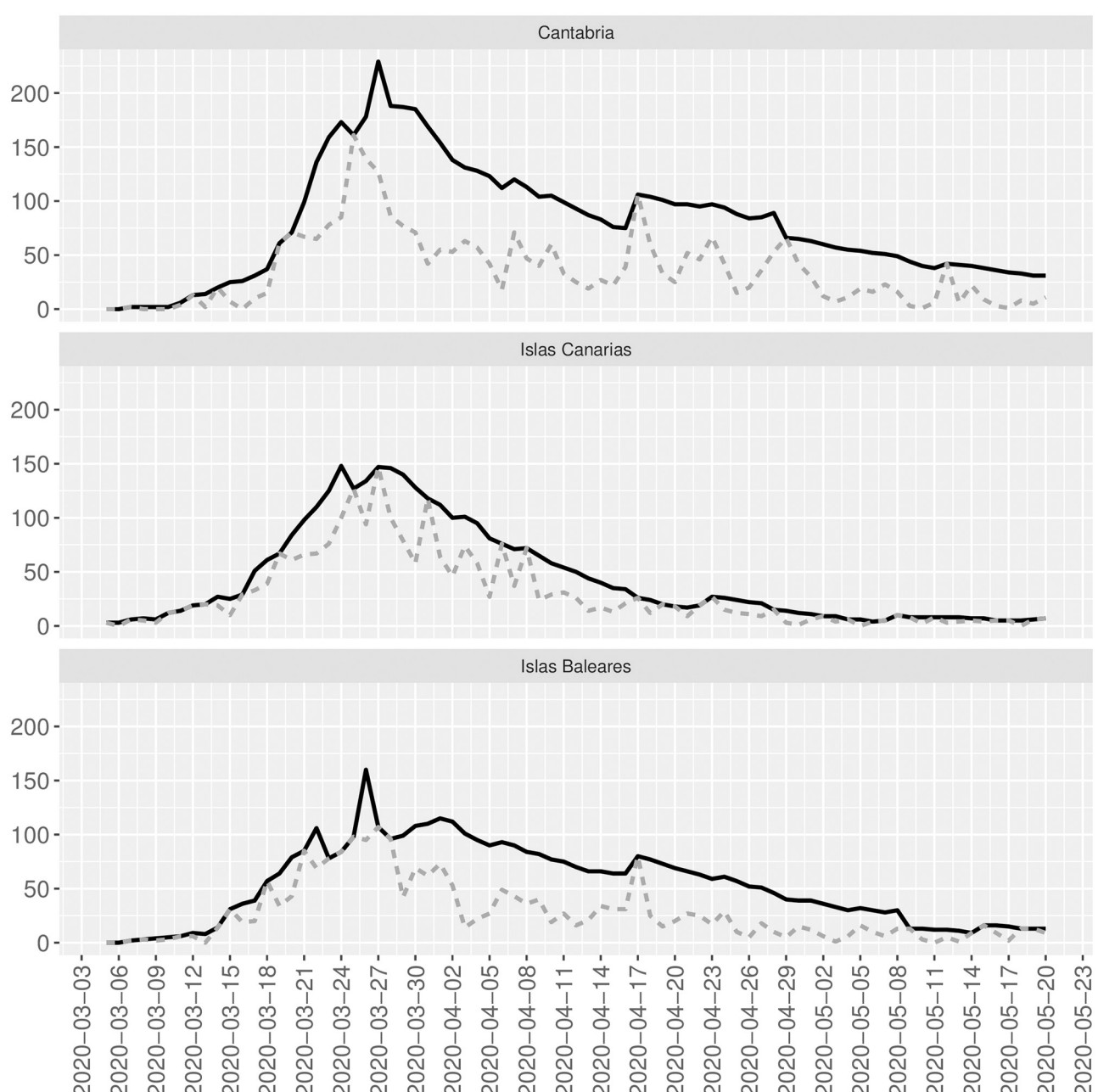

**Fig 1. Observed and reconstructed time series.** Gray-dotted lines are the observed time series for Cantabria (top), Islas Canarias (middle), and Islas Baleares (bottom). Black-bold lines are the reconstructed time series with the Viterbi algorithm.

Poisson distributed innovations $W_n$ is a function of time, $\lambda_n = f(\Theta, n)$, where $\Theta$ is a vector of parameters. Particularly, $\lambda_n = \text{new}(n) = A(n) - A(n-1)$, where $A(n)$ is defined by (7), assuming that $A_0 = 1$. The value of $A_0$ is known, representing the number of affected people at the starting time. However, if this value also needs to be estimated, it should thus be kept in the expression (7) as a parameter.

The model (4) defines the under-reported as an independent process in the sense that the state of under-reporting at time $n$ is not affected by the same state at time $n-1$. However, Fernández-Fontelo *et.at.* in [12] introduced a version of the under-reporting scheme according to

a two-state Markov chain. Although this approach of the under-reporting process could have been considered in the current model, the resulting model would be significantly more complicated than the one presented here from a computational perspective. In addition, the under-reporting process in (4) could also be considered stationary since it remains constant throughout the study (e.g., the under-reporting does not vary with time).

In the present work, however, the under-reporting process is flexible in that we do not restrict the under-reporting parameters to be constant over time; they can vary throughout the study if needed. To do so, both under-reporting parameters $\omega$ and $q$ can be made to vary with time, that is, $\omega_n = f(n, \Gamma)$ and $q_n = f(n, \Delta)$, where $\Gamma$ and $\Delta$ are vectors of parameters. In our current model, just the parameter $q$ is considered time-dependent, and not both parameters to reduce computational issues. The latter means that, if both parameters $\omega$ and $q$ are considered time-dependent, the resulting model is more complex and often shows convergence problems.

Particularly, the intensity of the under-reporting (i.e., $q$) is adjusted by the following logistic function:

$$q_n = \frac{\exp\left(\gamma_0 + \gamma_1 n + \gamma_2 \sin\left(\frac{2\pi n}{7}\right) + \gamma_3 \cos\left(\frac{2\pi n}{7}\right)\right)}{1 + \exp\left(\gamma_0 + \gamma_1 n + \gamma_2 \sin\left(\frac{2\pi n}{7}\right) + \gamma_3 \cos\left(\frac{2\pi n}{7}\right)\right)}, \tag{10}$$

Hence, we ensure that $q_n \in (0, 1)$. In expression (10), $\gamma_1$ indicates whether $q$ increases or decreases over time, while $\gamma_2$ and $\gamma_3$ indicate whether the series has a seasonal pattern with period $p = 7$ (weekly). Notice that if $\gamma_1 = \gamma_2 = \gamma_3 = 0$, then the previous logistic function becomes constant and thus $q_n = q$, resulting in the model (4). Hence, considering this function for the intensity of the under-reporting, the under-reporting process in the present model is defined by:

$$Y_n = \begin{cases} X_n & \text{with probability } 1 - \omega, \\ q_n \circ X_n & \text{with probability } \omega. \end{cases} \tag{11}$$

The parameters of the new model defined in (9) and (11) can be estimated using the forward algorithm through the forward probabilities defined in (5). In addition, the Viterbi algorithm introduced before can also be used to reconstruct the most likely latent sequences.

**Model forecasting.**   Another interesting point of the current analysis is the prediction of new daily cases of SARS-CoV2. These predictions can be used as a tool for foreseeing potential future outbreaks of the disease and, therefore, helping to implement earlier measures to lessen the impact of outbreaks. We propose two different predictors.

The most straightforward way to predict the values of $Y_{n+1}, Y_{n+2}, \ldots$ given the sample values $Y_1, Y_2, \cdots, Y_n$ is by considering their average point predictions, that is, $E(Y_{n+1})$, $E(Y_{n+2})$. . .. In particular, since the model (1) is an auto-regressive model of order 1, these average point predictions are expressed in terms of the last observed value, that is, $Y_n$.

According to the properties of the binomial thinning operator, we have that $E(Y_{n+k}) = E(X_{n+k})(1 - \omega(1 - q_n))$, where $E(X_{n+k}) = \frac{\lambda_{n+k}}{1-\alpha}$ and $\lambda_{n+k} = A(n + k) - A(n + k - 1)$. On the other hand, it is easy to see that $E(X_{n+k}) = \alpha^k E(X_n) + \sum_{i=1}^{k} \alpha^{k-i} \lambda_{n+i}$. Hence, if we have estimates for the corresponding parameters at a given time $n + k$, the average point prediction of $Y_{n+k}$ can be computed as follows:

$$E(Y_{n+k}) = \varphi_{n+k} \alpha^k Y_n + \varphi_{n+k}(1 - \omega(1 - q_n)) \sum_{i=1}^{k} \alpha^{k-i} \lambda_{n+i}, \tag{12}$$

where $\varphi_{n+k} = \frac{1-\omega(1-q_{n+k})}{1-\omega(1-q_n)}$. See S2 Appendix for more details on the computations.

The standard errors of these predictions (12) can be estimated using the Delta method. Briefly, the estimated variance of the prediction $\hat{E}(Y_{n+k})$, that is,

$\widehat{\text{Var}}(\hat{E}(Y_{n+k})) = \nabla \hat{E}(Y_{n+k})^T \Sigma \nabla \hat{E}(Y_{n+k})$, where $E(Y_{n+k})$ follows from (12), $\nabla E(Y_{n+k}) =$

$\left( \frac{\partial E(Y_{n+k})}{\partial \alpha}, \frac{\partial E(Y_{n+k})}{\partial m}, \frac{\partial E(Y_{n+k})}{\partial \beta}, \frac{\partial E(Y_{n+k})}{\partial \omega}, \frac{\partial E(Y_{n+k})}{\partial \gamma_0}, \frac{\partial E(Y_{n+k})}{\partial \gamma_1}, \frac{\partial E(Y_{n+k})}{\partial \gamma_2} \right)$ is the gradient function of $E(Y_{n+k})$, and $\Sigma$ is the variance-covariance matrix of the estimators of the parameters. Finally, the confidence intervals of $\hat{E}(Y_{n+k})$ can be easily computed as $\hat{E}(Y_{n+k}) \pm 1.96 \sqrt{\widehat{\text{Var}}(\hat{E}(Y_{n+k}))}$.

We can also predict an individual value of $Y_{n+k}$ based on its conditional distribution given the last value of the latent process $X_n$. This distribution is (see S3 Appendix):

$$Y_{n+k}|X_n = x_n \sim \begin{cases} \text{Binomial}(\alpha^k, x_n) + \text{Poisson}(\sum_{i=1}^{k} \alpha^{k-i} \lambda_{n+i}) & 1 - \omega, \\ \text{Binomial}(q_{n+k}\alpha^k, x_n) + \text{Poisson}(q_{n+k} \sum_{i=1}^{k} \alpha^{k-i} \lambda_{n+i}) & \omega. \end{cases} \quad (13)$$

The distribution (13) is a mixture of two components that are sums of a Binomial distribution and a Poisson distribution. To compute the corresponding probabilities for each component, a direct modification of expression (3) can be used. Finally, if $P_1(Y_{n+k} = j|X_n = x_n)$ is the probability of $Y_{n+k} = j$ in the first component of the mixture (13), and $P_2(Y_{n+k} = j|X_n = x_n)$ the same probability in the second component, the probability that $P(Y_{n+k} = j|X_n = x_n)$ of the mixture (13) is $P(Y_{n+k} = j|X_n = x_n) = (1 - \omega)P_1(Y_{n+k} = j|X_n = x_n) + \omega P_2(Y_{n+k} = j|X_n = x_n)$.

Given the distribution (13), and replacing the parameters by the maximum likelihood estimates, we can also estimate regions of prediction of size $1 - \alpha^*$ finding the lower and upper limits $r_1$ and $r_2$ that satisfy: $\sum_{j=1}^{r_1} P(Y_{n+k} = j|X_n = x_n) \approx \alpha^*/2$ and $\sum_{j=1}^{r_2} P(Y_{n+k} = j|X_n = x_n) \approx 1 - \alpha^*/2$.

## Results

The current application is based on the official daily number of confirmed SARS-CoV2 cases in different areas of Spain. In particular, it shows that the model presented before can be used to identify and quantify the under-reporting in small regions of Spain as well as in larger areas that can be officially and hierarchically divided into smaller regions (e.g., areas that can be divided into provinces or sanitary regions). That is, the model is ideal for quantifying the under-reporting issue locally and brings a solution to study that phenomenon in larger areas by aggregating the information in their smaller regions. Also, at the same time the under-reporting is estimated, the model accommodates the spread of the pandemic and provides this information through the parameters $M^*$ and $k$.

### Under-reporting of SARS-CoV2 in small areas of Spain

Three different small areas from Spain in the North (Cantabria), South (Islas Canarias), and Mediterranean coast (Islas Baleares) have been selected. The data from these areas consist of the number of confirmed cases by PCR tests. The day of confirmation coincides with the actual day the patient manifests symptomatology. See "Availability of data and codes" section for data availability. All time series range in the period from March 5 to May 20, 2020.

The time series corresponding to Cantabria takes values ranging from 0 to 161 cases per day, with a mean of 36 and 2788 positive PCR cases. The number of deaths is 209, 36 deaths per 100000 inhabitants since the beginning of the pandemic, set on February 20, 2020.

The time series for Islas Canarias takes values ranging from 0 to 147 positive PCR cases per day and with an average of 30 roughly and a total of 2299 positive cases by PCR. A total of 155 people died, which means seven deaths per 100000 inhabitants since the beginning of the pandemic.

The time series for Islas Baleares has values ranging from 0 to 107, with an average of 28 and 2125 positive cases by PCR. Two hundred twenty-one deaths are registered in this area since the beginning of the pandemic. This implies a total of 19 deaths per 100000 inhabitants.

Fig 1 shows the evolution over time of the new daily positive cases by PCR in Cantabria (top), Islas Canarias (middle), and Islas Baleares (bottom). The graph shows that these time series are governed by a trend that increases to a maximum peak (the peak of the pandemic) and decreases. Therefore, it is evident that the time series are non-stationary. Additionally, the series shows periodic peaks that coincide with the "weekend effect" previously described.

Table 1 shows the maximum likelihood estimates (MLE) of the model defined in (9) and (11). For Cantabria, the overall frequency of under-reporting, that is, $\omega$ is estimated as $\hat{\omega} = 0.8814$ and the intensity, that follows the function (10), is $\hat{q}_n = \frac{\exp\left(0.3875 - 0.0197n + 0.3203\sin\left(\frac{2\pi n}{7}\right) + 0.1748\cos\left(\frac{2\pi n}{7}\right)\right)}{1 + \exp\left(0.3875 - 0.0197n + 0.3203\sin\left(\frac{2\pi n}{7}\right) + 0.1748\cos\left(\frac{2\pi n}{7}\right)\right)}$.

On the other hand, the latent process for Cantabria is estimated as $X_n = 0.9653 \circ X_{n-1} + W(\hat{\lambda}_n)$, where $\hat{\lambda}_n = \hat{A}(n) - \hat{A}(n-1)$ and $\hat{A}(n) = \frac{237.99e^{0.3304n}}{237.99 + e^{0.3304n} - 1}$. For the other two regions, the models are similar to the model for it Cantabria. In particular, for Islas Canarias, the overall frequency and intensity of the under-reporting process are $\hat{\omega} = 0.7943$ and $\hat{q}_n = \frac{\exp\left(0.9469 - 0.0218n + 0.2313\sin\left(\frac{2\pi n}{7}\right) - 0.0570\cos\left(\frac{2\pi n}{7}\right)\right)}{1 + \exp\left(0.9469 - 0.0218n + 0.2313\sin\left(\frac{2\pi n}{7}\right) - 0.0570\cos\left(\frac{2\pi n}{7}\right)\right)}$. The latent process for Islas Canarias is estimated as $X_n = 0.9271 \circ X_{n-1} + W(\hat{\lambda}_n)$ where $\hat{A}(n) = \frac{256e^{0.3271n}}{256 + e^{0.3271n} - 1}$. Finally, for Islas Baleares, the under-reporting process parameters' are estimated as $\hat{\omega} = 0.7913$ and

**Table 1. MLE for Cantabria, Islas Canarias and Islas Baleares.**

|  | Cantabria | Islas Canarias | Islas Baleares |
|---|---|---|---|
| $\hat{\alpha}$ | 0.9653 | 0.9271 | 0.9539 |
| s.e.$\hat{\alpha}$ | 0.0034 | 0.0066 | 0.0046 |
| $\hat{M}^*$ | 237.99 | 256.00 | 194.69 |
| s.e.$_{\hat{M}^*}$ | 18.99 | 22.54 | 17.77 |
| $\hat{k}$ | 0.3304 | 0.3271 | 0.2919 |
| s.e.$\hat{k}$ | 0.0141 | 0.0092 | 0.0107 |
| $\hat{\omega}$ | 0.8814 | 0.7943 | 0.7913 |
| s.e.$\hat{\omega}$ | 0.0406 | 0.0521 | 0.0496 |
| $\hat{\gamma}_0$ | 0.3875 | 0.9469 | 0.3485 |
| s.e.$_{\hat{\gamma}_0}$ | 0.1071 | 0.1289 | 0.1386 |
| $\hat{\gamma}_1$ | -0.0197 | -0.0218 | -0.0232 |
| s.e.$_{\hat{\gamma}_1}$ | 0.0024 | 0.0037 | 0.0037 |
| $\hat{\gamma}_2$ | 0.3203 | 0.2313 | -0.1004 |
| s.e.$_{\hat{\gamma}_2}$ | 0.0454 | 0.0629 | 0.0540 |
| $\hat{\gamma}_3$ | 0.1748 | -0.0570 | 0.4145 |
| s.e.$_{\hat{\gamma}_3}$ | 0.0432 | 0.0618 | 0.0556 |

Table gives the MLE and standard errors of the parameters of the model defined in (9) and (11) for Cantabria, Islas Canarias and Islas Baleares.

$\hat{q}_n = \frac{\exp\left(0.3485 - 0.0232n - 0.1004\sin\left(\frac{2\pi n}{7}\right) + 0.4145\cos\left(\frac{2\pi n}{7}\right)\right)}{1 + \exp\left(0.3485 - 0.0232n - 0.1004\sin\left(\frac{2\pi n}{7}\right) + 0.4145\cos\left(\frac{2\pi n}{7}\right)\right)}$. The latent process for Islas Baleares is $X_n = 0.9539 \circ X_{n-1} + W(\hat{\lambda}_n)$ where $\hat{A}(n) = \frac{194.69 e^{0.2919n}}{194.69 + e^{0.2919n} - 1}$.

Fig 1 shows the officially registered new daily SARS-CoV2 cases confirmed by PCR in Cantabria, Islas Canarias, and Islas Baleares from 5 March to 20 May (grey lines). The graph also shows the reconstructed time series with the Viterbi algorithm for the above areas (black lines). Although the Spanish authorities have confirmed by PCR a total of 2788, 2299, and 2125 new SARS-CoV2 cases in the studied period in Cantabria, Islas Canarias, and Islas Baleares, the model presented here estimates a total of 6074, 3370, and 4079 new cases within this period in the areas mentioned above. That is, officially Cantabria, Islas Canarias, and Islas Baleares, only the 45.9%, 68.2%, and 52.9% of the total new SARS-CoV2 cases by PCR are registered.

On 20 May, 209, 155, and 221 people died due to SARS-CoV2 in Cantabria, Islas Canarias, and Islas Baleares, respectively. As expected, the lethality rates computed using the observed and reconstructed number of confirmed cases by PCR differ. While these rates are 7.5%, 6.7%, and 10.4% in Cantabria, Islas Canarias, and Islas Baleares, the reconstructed rates significantly decrease to 3.4%, 4.6%, and 5.4%.

Results in Table 1 also allow reconstructing the SIR model, and therefore estimating the parameters $\beta$, $\gamma$, and $N$, also using the number of affected people $A^*$ when the curve $A(t)$ grows fastest (see S1 Appendix).

Although the SIR model's exact solution can be derived, in our model, an approximate solution to the SIR model has been considered the logistic function $A(t)$ to make the model computationally less expensive. Because our SIR estimation relies on an approximated solution, in practice, in some cases, the reconstruction of the parameters $\beta$, $\gamma$, and $N$ is not possible since the equation (S1.11) has no proper solution.

In the case of Cantabria, Islas Canarias and Islas Baleares, a proper solution for (S1.11) has been found for the three regions. In particular, for Cantabria, considering the estimated parameters $\hat{\alpha} = 0.9653$, $\hat{M}^* = 237.99$, $\hat{k} = 0.3344$ and observing that the fastest growth of $A(t)$ occurs at $A^* = 1631.9$, we obtain $A_\infty \approx 6858.5$, $A_0 = 1/(1 - \hat{\alpha}) = 28.8$ and, solving numerically (S1.11), $\hat{N} = 427418.7$. Then, plugging the value of $\hat{N}$ in (8) and using that $\hat{\beta} - \hat{\gamma} = 0.3344$ we find $\hat{\gamma} = 85.57$ and $\hat{\beta} = 85.90$.

Acting similarly for the other two regions, for Islas Canarias $\hat{N} = 50629.4$, $\hat{\gamma} = 10.07$ and $\hat{\beta} = 10.40$. For Islas Baleares, $\hat{N} = 79584.6$, $\hat{\gamma} = 13.05$ and $\hat{\beta} = 13.34$

Fig 2 shows the forecasting results based on the average point prediction and the $k$-ahead forecasting distribution (e.g., percentiles 2.5%, 50%, and 97%) for the areas mentioned above using a dynamic and static approach. The dynamic method is usually used to evaluate the models' predictive capability and consists of splitting down the time series into the training and testing time series sets of sizes $n - k$ and $k$. The method starts training the model over the $n - k$ observation in the training set. The prediction and the 95% confidence levels for the observation $n - k + 1$ are computed through the trained model and compared to the true observation $n - k + 1$. After that, the training set is updated by including the first $n - k + 1$, the model is re-fitted over the new training set, and a new prediction for the observation $n - k + 2$ is computed. This recursive process is repeated until the last prediction for the $n$ observation is computed over the training set containing the $n - 1$ first observations. The static method is usually used to predict unknown future values. The idea consists of using the last $X_n$ value to predict both the average point prediction at time $t + k$ and the $k$-ahead forecasting distribution.

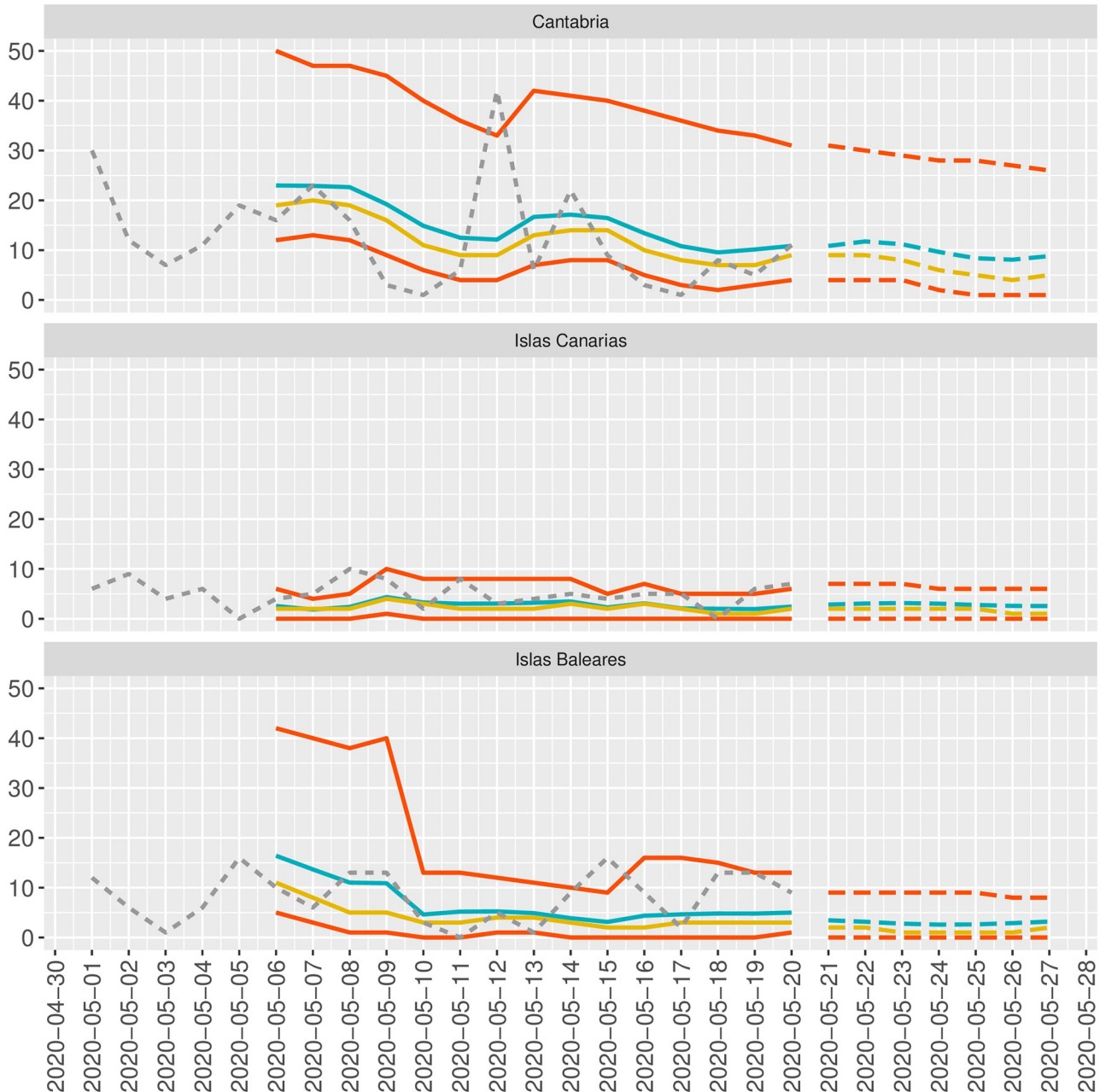

**Fig 2. Dynamic and static forecasting.** Dynamic (solid lines) and static (dotted lines) forecasting for Cantabria, Islas Canarias and Islas Baleares. Red lines correspond to the percentiles 2.5% and 97.5%, blue line corresponds to mean, and yellow line corresponds to median.

Fig 2 shows the dynamic prediction from 6 May to 20 May. In particular, the graph shows the average point prediction (blue line), the median point prediction (yellow lines), and the percentiles 2.5% and 97.5% (red-solid lines) of the forecasting distribution. For all areas, it is clear that most of the time, the observed values are within the percentiles 2.5% and 97.5%, that is, within the 95% confidence levels. This figure also shows the static prediction from 21 May to 27 May, a period where no observations are available (supplemental material).

## Under-reporting of SARS-CoV2 in large areas of Spain

In this second example, the number of daily SARS-CoV2 cases confirmed by PCR is studied in two large areas of Spain by splitting these areas into smaller hierarchical regions (e.g., areas divided into smaller areas according to geographical or sanitary reasons). In these cases, as before, the day of confirmation coincides with the actual day the patient had symptoms. Recall that the model introduced here is especially useful for studying the evolution of the SARS-CoV2 cases and identifying and quantifying its under-reported in small areas. However, in this example, we will show that larger areas can also be studied with these models if we have a way to split these vast areas.

The autonomous community of Galicia is divided into seven different sanitary areas. These areas' data consist of the number of daily new cases of SARS-CoV2 confirmed by PCR from 12 March to 27 April. For the Galician data, the series had to be cut on 28 April since the region's health system changed the definition of new cases from 28 April onwards. Overall the autonomous community, the minimum and the maximum number of new daily cases confirmed by PCR range from 0 to 185, with 21 cases per day on average. A total of 6974 cases are registered in Galicia as confirmed cases by PCR on 28 April. See "Availability of data and codes" section for data availability.

The autonomous community of Andalucía is divided into seven provinces. In this case, each province's time series is the number of new daily SARS-CoV2 cases confirmed by PCR from 5 March to 20 May. Overall Andalucía, the minimum and the maximum number of daily cases confirmed by PCR range from 0 to 185, with 20.4 cases per day on average. A total of 12591 cases are officially registered in Andalucía as confirmed cases by PCR on 20 May. See "Availability of data and codes" section for data availability.

Table 2 shows the maximum likelihood estimates of the model in (9) and (11), or nested versions, for each of the sanitary areas and provinces of Galicia and Andalucía, respectively. It can be seen in that table that the models between lower regions within the same area are consistent.

Figs 3 and 4 show the time series for each sanitary area or provinces and the corresponding reconstructions for Galicia and Andalucía, respectively.

For Galicia, from 12 March to 27 April 1630, 1269, 1096, 577, 1323, 670, and 409 cases of SARS-CoV2 are officially registered in Coruña, Vigo, Santiago, Pontevedra, Ourense, Lugo, and Ferrol, respectively. However, our model estimates that, over the same period previously defined, 3559, 3062, 2112, 951, 3922, 1363, and 1121 cases of SARS-CoV2 in with the same characteristics and the corresponding areas above really occurred. The latter implies that only 45.8%, 41.4%, 51.89%, 60.7%, 33.7%, 49.2% and 36.5% of the total registered cases of SARS-CoV2 that have been confirmed by PCR has been observed in Coruña, Vigo, Santiago, Pontevedra, Ourense, Lugo, and Ferrol, respectively. Overall in Galicia, a total of 6974 cases are registered between 12 March to 27 April. Our model estimates that the actual number of cases with the same characteristics is 16090; that is, only 43.3% of the cases have been officially registered. During that period, 405 people died due to SARS-CoV2 that implies a lethality of 5.8% or 2.5% depending on whether the denominator is the observed or reconstructed total cases, respectively. The mortality rate overall Galicia is estimated as 15 deaths per 100000 inhabitants.

The estimation of the parameters of the SIR model can be obtained as described in the first example. For instance, for Pontevedra $\hat{N} = 5129.6$, $\hat{\gamma} = 2.76$ and $\hat{\beta} = 3.02$, and for Ferrol $\hat{N} = 55790.5$, $\hat{\gamma} = 39.83$ and $\hat{\beta} = 40.11$.

For Andalucia, from 5 March to 20 May, 497, 1252, 1338, 2437, 401, 1443, 2761, and 2462 cases of SARS-CoV2 are officially registered in Almería, Cádiz, Córdoba, Granada, Huelva,

**Table 2. MLE for Galicia and Andalucía.**

| | Galicia | | | | | | | |
| --- | --- | --- | --- | --- | --- | --- | --- | --- |
| | Coruña | Vigo | Santiago | Pontevedra | Ourense | Lugo | Ferrol | |
| $\hat{\alpha}$ | 0.9391 | 0.8929 | 0.7661 | 0.8813 | 0.9232 | 0.6115 | 0.7145 | - |
| s.e.$\hat{\alpha}$ | 0.0055 | 0.0090 | 0.0241 | 0.0211 | 0.0075 | 0.0375 | 0.0403 | - |
| $\hat{M}$ | 266.57 | 342.37 | 468.99 | 107.66 | 329.61 | 527.08 | 322.85 | - |
| s.e.$\hat{M}$ | 21.51 | 28.03 | 50.50 | 19.31 | 27.57 | 55.48 | 44.93 | - |
| $\hat{k}$ | 0.3744 | 0.3695 | 0.3052 | 0.2519 | 0.3526 | 0.2856 | 0.2858 | - |
| s.e.$\hat{k}$ | 0.0102 | 0.0087 | 0.0067 | 0.0171 | 0.0098 | 0.0061 | 0.0075 | - |
| $\hat{\omega}$ | 0.7500 | 0.7591 | 0.6264 | 0.6787 | 0.8891 | 0.5467 | 0.7200 | - |
| s.e.$\hat{\omega}$ | 0.0625 | 0.0632 | 0.0717 | 0.0739 | 0.0550 | 0.0823 | 0.0717 | - |
| $\hat{\gamma}_0$ | -0.9697 | 0.3823 | 0.2133 | 1.7702 | 0.3805 | 0.1192 | -1.4025 | - |
| s.e.$_{\hat{\gamma}_0}$ | 0.0483 | 0.1587 | 0.2353 | 0.3156 | 0.1581 | 0.5829 | 0.1095 | - |
| $\hat{\gamma}_1$ | - | -0.0672 | -0.0494 | -0.0987 | -0.0527 | -0.0979 | - | - |
| s.e.$_{\hat{\gamma}_1}$ | - | 0.0068 | 0.0098 | 0.0121 | 0.0059 | 0.0225 | - | - |
| $\hat{\gamma}_2$ | 0.2369 | - | -0.3834 | -0.4469 | 0.1610 | 0.6420 | 0.4011 | - |
| s.e.$_{\hat{\gamma}_2}$ | 0.0580 | - | 0.1009 | 0.1591 | 0.0566 | 0.1714 | 0.1316 | - |
| $\hat{\gamma}_3$ | 0.0250 | - | -0.0778 | 0.1760 | -0.3460 | -1.0377 | -0.1089 | - |
| s.e.$_{\hat{\gamma}_3}$ | 0.0550 | - | 0.0937 | 0.1512 | 0.0582 | 0.2606 | 0.1309 | - |
| | Andalucia | | | | | | | |
| | Almería | Cádiz | Córdoba | Granada | Huelva | Jaén | Málaga | Sevilla |
| $\hat{\alpha}$ | 0.9198 | 0.9188 | 0.8691 | 0.9240 | 0.7608 | 0.9289 | 0.9030 | 0.9212 |
| s.e.$\hat{\alpha}$ | 0.0154 | 0.0098 | 0.0151 | 0.0073 | 0.0582 | 0.0084 | 0.0077 | 0.0070 |
| $\hat{M}$ | 77.22 | 163.15 | 260.28 | 271.67 | 155.54 | 190.86 | 354.23 | 312.86 |
| s.e.$\hat{M}$ | 13.38 | 18.32 | 29.18 | 24.56 | 35.58 | 21.29 | 28.00 | 25.61 |
| $\hat{k}$ | 0.2438 | 0.2879 | 0.2626 | 0.3226 | 0.2161 | 0.2620 | 0.3608 | 0.3310 |
| s.e.$\hat{k}$ | 0.0163 | 0.0110 | 0.0080 | 0.0093 | 0.0093 | 0.0092 | 0.0101 | 0.0080 |
| $\hat{\omega}$ | 0.8400 | 0.8306 | 0.6864 | 0.8151 | 0.8114 | 0.7910 | 0.7717 | 0.9011 |
| s.e.$\hat{\omega}$ | 0.0537 | 0.0493 | 0.0601 | 0.0492 | 0.0713 | 0.0491 | 0.0514 | 0.0365 |
| $\hat{\gamma}_0$ | 1.1195 | 1.3226 | 0.1061 | 1.3984 | -0.0319 | 0.7684 | 1.1918 | 1.5830 |
| s.e.$_{\hat{\gamma}_0}$ | 0.3079 | 0.1925 | 0.2020 | 0.1391 | 0.1490 | 0.1445 | 0.1363 | 0.1386 |
| $\hat{\gamma}_1$ | -0.0432 | -0.0352 | -0.0144 | -0.0337 | -0.0160 | - | -0.0242 | -0.0392 |
| s.e.$_{\hat{\gamma}_1}$ | 0.0093 | 0.0048 | 0.0062 | 0.0039 | 0.0149 | - | 0.0039 | 0.0040 |
| $\hat{\gamma}_2$ | -0.2270 | 0.2430 | 0.4398 | 0.1198 | 0.1489 | 0.2477 | 0.1004 | 0.1734 |
| s.e.$_{\hat{\gamma}_2}$ | 0.1326 | 0.0979 | 0.0897 | 0.0736 | 0.1636 | 0.1600 | 0.0636 | 0.0602 |
| $\hat{\gamma}_3$ | 0.5452 | 0.2660 | -0.0627 | 0.4262 | 0.6838 | 0.3973 | 0.3880 | 0.4711 |
| s.e.$_{\hat{\gamma}_3}$ | 0.1280 | 0.0871 | 0.1001 | 0.0658 | 0.1660 | 0.0748 | 0.0790 | 0.0594 |

Tables gives the MLE and standard errors of the parameters of the model defined in (9) and (11) for Galicia and Andalucía

Jaén, Málaga, and Sevilla, respectively. However, our models estimate 856, 1908, 2025, 3525, 644, 2552, 3845, and 3847 over the same period and respectively, for the same areas. That is, only 58.1%, 65.6%, 66.1%, 69.1%, 62.3%, 56.5%, 71.8% and 64.0% of the total registered cases of SARS-CoV2 that have been confirmed by PCR has been observed in Almería, Cádiz, Córdoba, Granada, Huelva, Jaén, Málaga, and Sevilla, respectively. Overall Andalucía, a total of 12591 cases are registered from 5 March to 20 May. Our model estimates a total of 19202 cases

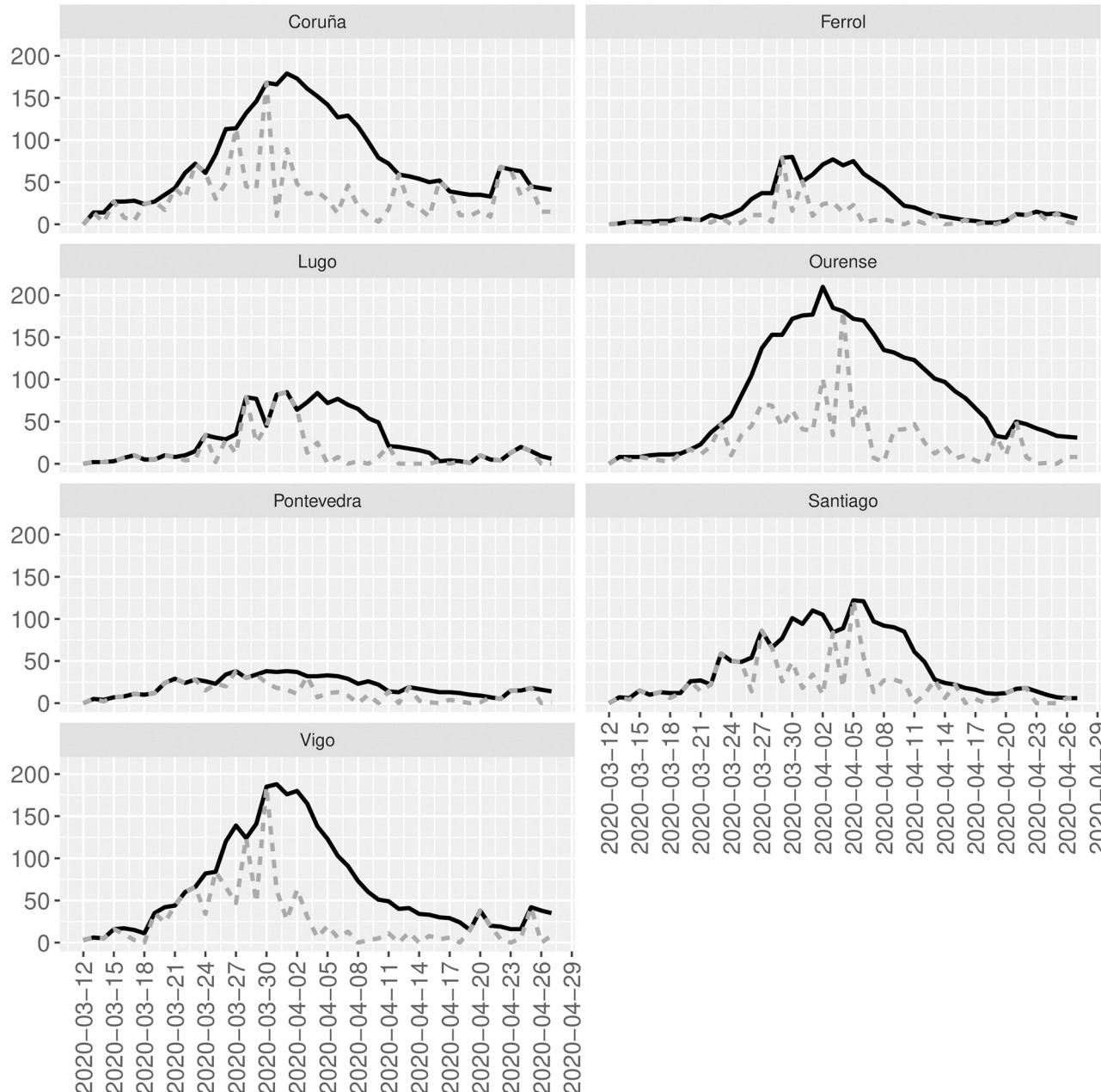

**Fig 3. Observed and reconstructed time series.** Gray-dotted lines are the observed time series for each sanitary area of Galicia. Black-bold lines are the reconstructed time series for each sanitary area of Galicia.

with the same characteristics as those in the registered cases; that is, only 65.6% are registered in this autonomous community. As before, the lethality rate strongly differs depending on whether the denominator is considered as the observed or reconstructed total of cases over the specified period. In particular, overall Andalucia, 1371 people died over the specified period, which implies lethality rates of 10.9% or 7.1% if the number of total cases corresponds to the officially registered or reconstructed, respectively. The mortality rate overall Andalucía is estimated as 16 deaths per 100000 inhabitants.

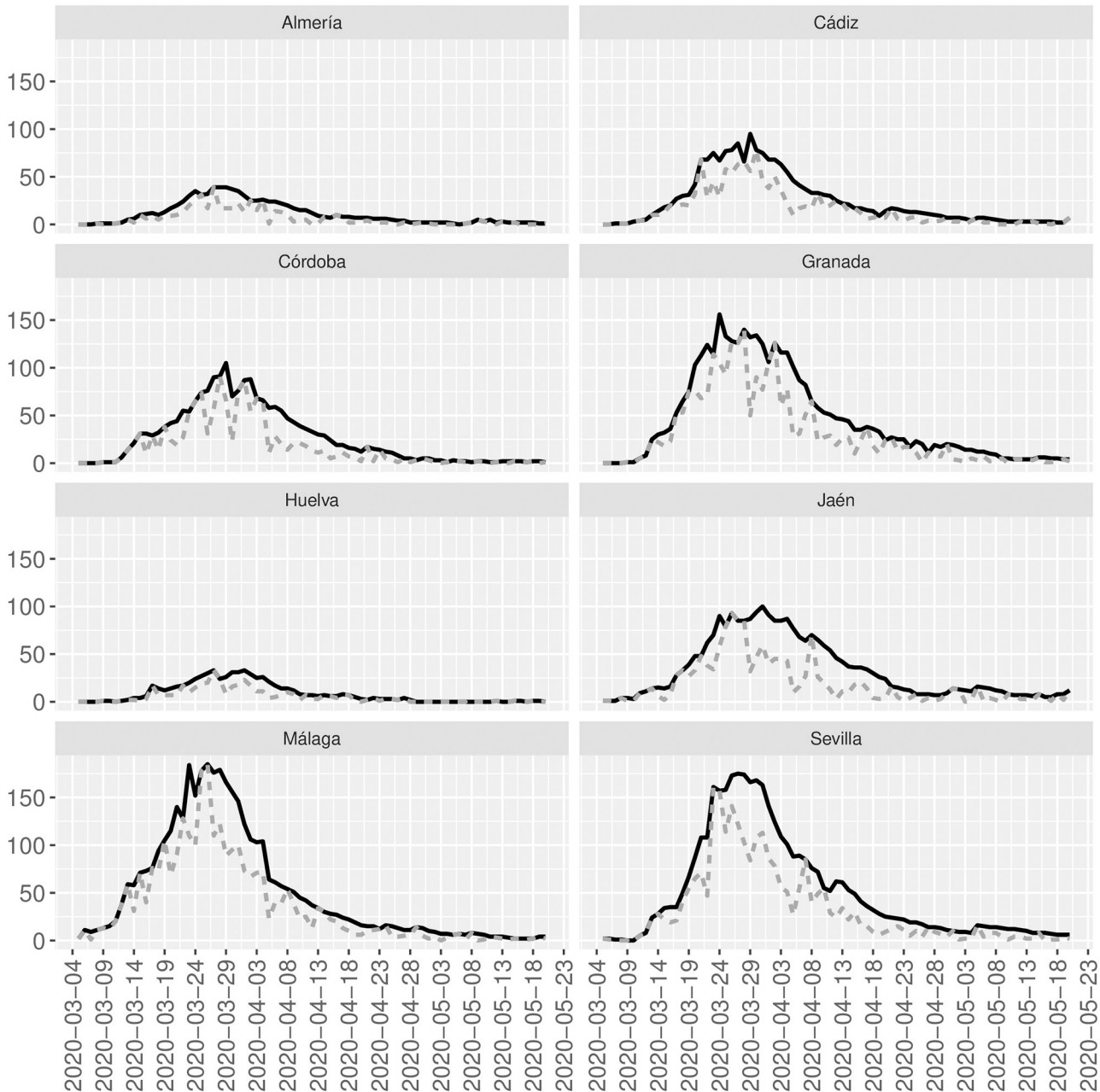

**Fig 4. Observed and reconstructed time series.** Gray-dotted lines are the observed time series for each province of Andalucía. Black-bold lines are the reconstructed time series for each province of Andalucía.

Concerning the reconstruction of the SIR models, for example, for Córdoba $\hat{N} = 22500.2$, $\hat{\gamma} = 6.06$ and $\hat{\beta} = 6.33$, and for Huelva $\hat{N} = 7842.1$, $\hat{\gamma} = 5.77$ and $\hat{\beta} = 5.98$.

Fig 2 can also be built for both the areas of Galicia and Andalucía in the same way than what we did before for Cantabria, Islas Canarias and Islas Baleares.

## Discussion

One of the major challenges in the struggle against the SARS-CoV2 pandemic relies on the people who come down with a mild form of the disease (e.g., experiencing mild symptoms or

even being asymptomatic) and therefore constitutes one of the most powerful vectors of virus' transmission, combined with the lack of tests that impede carrying out large-scale screenings [1]. However, the quick and efficient identification of those people is vital to earlier control potential trends of infection and evaluate the pandemic's impact (e.g., to get unbiased estimators on the spread and impact of the epidemic).

Since the epidemic showed up last December in China, the concern with the under-reporting in official data as the number of infections and deaths worldwide has been on everyone's lips, including the media [34–36].

With this in mind, this paper aims to introduce an extension of the model proposed in [11] to estimate the magnitude of the under-reporting in epidemics such as the current SARS-CoV2. That article presents a model that considers two processes: an underlying process (true process) that we do not observe, and the observed and potentially under-reported process that provides a proportion of what happens (a proportion of the true process). The model's particularity is that the underlying process assumes an INAR(1) structure, and hence a particular correlation structure. The model measures the under-reporting with two parameters that estimate the number of times the process is under-reported ($\omega$), and the overall distance between the most likely sequence of latent states and the observed sequence ($q$). However, the model in [11] is intended to fit a stationary time series, which is not the case of the SARS-CoV2 data. Therefore, to adapt the model above to the SARS-CoV2 case, the underlying process's expected value is allowed to be time-dependent through an approximated solution of the SIR differential equations that depend on the new affected individuals at each time. The new version of the model also allows considering time-varying under-reporting, which, in the SARS-CoV2 case, may sometimes be more realistic (e.g., the intensity of the under-reporting may decrease if the number of large-scale screenings increases). Thus, the resulting model measures the under-reporting and adapts the pandemic's evolution based on the SIR model at the same time.

This paper's results confirm that the under-reporting is effectively present in SARS-CoV2 data from various regions in Spain conditioned to different management, policies, and climate conditions. Results also show that the model has powerful predictive characteristics exhibited in Fig 2 and that the SIR parameters $N$, $\beta$, and $\gamma$ can be relatively quickly recovered from the results in Tables 1 and 2. As expected, the under-reporting from almost all regions is not constant over time but varies with times showing a decreasing trend ($\hat{\gamma}_1$) and a seasonal pattern with seven days of periodicity ($\hat{\gamma}_2$ and $\hat{\gamma}_3$). A decreasing trend means that the $q$ parameter tends to 0 as time increases, and therefore the intensity of the under-reporting phenomenon is stronger as time passes. This result is surprising since it was expected a less intense under-reporting process as time increases. However, the changes in protocols, data collection strategies, among others, could have affected the evolution of this under-reporting process. On the other hand, the seven-days periodicity is explained by the "weekend effect" that produces a systematic decrease in the number of new cases during the weekends.

It has been shown that the coverage percentages vary from 33.7% (Ourense in Galicia) to 71.8% (Málaga in Andalucía) and that the estimated lethality rates decrease significantly when the number of reconstructed cases, rather than the number based only on officially reported cases, are considered as the total number of affected individuals. In particular, lethality rates with official cases range from 5.8% to 10.9% and decrease to 2.5% and 7.1% with the reconstructed cases. The example with the lethality rates reveals the under-reporting influence on the parameters' estimates, which are often used to make decisions. Thus, the importance of having appropriate methods to identify, control, and estimate under-reporting.

Besides the under-reporting quantification, the model allows estimating the SIR model under the underlying process. In particular, for Cantabria, Islas Canarias, and Islas Baleares is

estimated that the number of susceptible people ($\hat{N}$) is 427418.7, 50629.4, and 79584.6, while the infection and removal rates (scaled by $\hat{N}$) are 0.0201% and 0.0200%, 0.0205% and 0.0199%, and 0.0168% and 0.0164%, respectively. The latter means the basic reproduction numbers ($\hat{\beta}/\hat{\gamma}$) are slightly over 1, which means that the virus, although nearly to be controlled, is not under control yet. For the second example, which is mainly intended to show how the model can be used for large areas with large counts, similar numbers have been obtained for $\hat{\beta}$ and $\hat{\gamma}$.

One of the current work's main limitations is based on the available data since the model can only estimate the unobserved counts that have similar characteristics to the observed data. Since our data do not contain asymptomatic, our estimated reconstructions in Figs 1, 3 and 4 do not contain those asymptomatic and only people with the same characteristics than the observed counts who have not been officially registered. To also include asymptomatic in the reconstruction of the underlying process, random tests should be done to include cases that have passed the infection asymptomatically. However, to the authors' knowledge, this information is not available yet. Other limitations are related to computational issues, especially when dealing with relatively large counts, and the sensitivity of the SIR model's approximated solution that sometimes does not allow recovering the parameters $\hat{N}$, $\hat{\beta}$, and $\hat{\gamma}$.

The model presented here constitutes a first approach to a reliable method to estimate a pandemic's under-reporting, such as the current SARS-CoV2. Furthermore, the model can be extended in different ways, such as considering more complex correlation structures in the underlying process (e.g., INAR(p) or INARMA model), or considering more general thinning operators for representing the observed process.

## Supporting information

**S1 Appendix. Detailed computations concerning the SIR model.**
(PDF)

**S2 Appendix. Detailed computations concerning the average predictions.**
(PDF)

**S3 Appendix. Detailed computations concerning the *k*-ahead forecasting distribution.**
(PDF)

## Acknowledgments

The authors would like to thank Gustavo Eduardo Ávalos Villaseñor for his help in data scrapping and processing.

## Author Contributions

**Conceptualization:** David Moriña, Alejandra Cabaña, Argimiro Arratia.

**Data curation:** Amanda Fernández-Fontelo, Alejandra Cabaña, Argimiro Arratia.

**Formal analysis:** Amanda Fernández-Fontelo, Alejandra Cabaña.

**Funding acquisition:** David Moriña.

**Investigation:** Pere Puig.

**Methodology:** Pere Puig.

**Project administration:** David Moriña.

**Software:** Argimiro Arratia.

**Writing – original draft:** Amanda Fernández-Fontelo.

**Writing – review & editing:** Amanda Fernández-Fontelo, David Moriña, Alejandra Cabaña, Argimiro Arratia, Pere Puig.

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
