## [Editor Report · Decision Letter 0]

7 Aug 2020

PONE-D-20-20748

Estimating the real burden of disease under a pandemic situation: The SARS-CoV2 case

PLOS ONE

Dear Dr. Fernández-Fontelo,

Thank you for submitting your manuscript to PLOS ONE. After careful consideration, we feel that it has merit but does not fully meet PLOS ONE’s publication criteria as it currently stands. Therefore, we invite you to submit a revised version of the manuscript that addresses the points raised during the review process.

We look forward to receiving your revised manuscript.

Kind regards,

Paul K. Newton, Ph.D.

Academic Editor

PLOS ONE

Additional Editor Comments:

We have had trouble getting reviewers to respond so I have read the paper carefully myself and am quite positive about it. The authors do a thorough job of focusing in on a specific question associated with the current pandemic - underreporting - and model this using nice data sets. The paper is well written and very clear in its presentation. I was not able to find any figure captions - maybe this has something to do with the electronic submission? If not, please provide figure captions, but otherwise I recommend publication.
---

## [Author Response · Author response to Decision Letter 0]

15 Sep 2020

Response to the Editor Comments 

Manuscript: Estimating the real burden of disease under a pandemic situation: the SARS-CoV2 case

 ID: PONE-D-20-20748

First of all, we would like to thank the Editor for the positive response regarding our paper. The Editor reported a unique minor change to our manuscript related to the captions of the Figures. The manuscript has four Figures that have been created as independent files and attached separately in the online submission system following the Guidelines. The captions of each of these figures has been incorporated in the manuscript (and highlighted in red) immediately before the corresponding figure is first cited in the text. In particular, the captions are the following ones: 

(page 9/18)

Fig 1. Observed and reconstructed time series.

Gray-dotted lines are the observed time series for Cantabria (top), Islas Canarias (middle), and Islas Baleares (bottom). Black-bold lines are the reconstructed time series with the Viterbi algorithm.

(page 10/18)

Fig 2. Dynamic and static forecasting.

Dynamic (solid lines) and static (dotted lines) forecasting for Cantabria, Islas Canarias and Islas Baleares. Red lines correspond to the percentiles 2.5% and 97.5%, blue line corresponds to the mean, and yellow line corresponds to the median.

(page 13/18)

Fig 3. Observed and reconstructed time series. 

Gray-dotted lines are the observed time series for each sanitary area of Galicia. Black-bold lines are the reconstructed time series for each sanitary area of Galicia. 

(page 13/18)

Fig 4. Observed and reconstructed time series.

Gray-dotted lines are the observed time series for each province of Andalucía. Black-bold lines are the reconstructed time series for each province of Andalucía.

---

## [Decision Letter · Decision Letter 1]

20 Oct 2020

PONE-D-20-20748R1

Estimating the real burden of disease under a pandemic situation: The SARS-CoV2 case

PLOS ONE

Dear Dr. Fernández-Fontelo,

Thank you for submitting your manuscript to PLOS ONE. After careful consideration, we feel that it has merit but does not fully meet PLOS ONE’s publication criteria as it currently stands. Therefore, we invite you to submit a revised version of the manuscript that addresses the points raised during the review process. See referee report.

Please submit your revised manuscript in one months timeframe if possible. If you will need more time than this to complete your revisions, please reply to this message or contact the journal office at plosone@plos.org. Please include the following items when submitting your revised manuscript:

We look forward to receiving your revised manuscript.

Kind regards,

Paul K. Newton, Ph.D.

Academic Editor

PLOS ONE

Additional Editor Comments (if provided):

This manuscript introduces a new observation model that accounts for underreporting of SARS-CoV2 cases by combining an SIR model with a stochastic observation model. In general the manuscript is clear and well written and the methods used are clearly described. I have two main comments though, the first one a technical concern, while the second one has to do with framing of the research.

Firstly the proposed model seems to allow for double counting of infections to occur, because tested individuals are not taken out of the number of real cases (Xn). This is especially a concern on days that the number of counted cases is (almost) as high as the number of actual cases, because on those days none of the cases should carry over to the next day. In that light alpha also seems extremely high, with in many locations, more than 90% of the cases being carried over from one day to the next, which could easily lead to cases being counted many times. Related to this the underreporting rate (1-q) also seems low (below 0.5 in many cases). Especially compared to other estimates of the ascertainment rate (e.g. 0.23 [3]). A low underreporting rate with a high alpha would lead to cases being double counted extremely often.

With regard to the framing of the research. In essence, the work presented here is seems to be about fitting a (simplified) SIR model to the outbreak, using an observation model. This is already an extremely rich field (e.g. 1-4) and many of those models do use observation models of various complexity. As far as I know the presented observation model is new in that it carries over cases from previous days using a stochastic process and the observation probability changes with the day of the week (q). Still I believe the work would be strengthened by a comparison with such models.

Minor comments:

- The logistic function should be represented with logit^-1 not logit (equation 10 and throughout the text)

- In the first supplementary material, equation labels are inconsistent. The equations are labelled using S1.x, while they are referred to (in the text) as A.x.

References:

[1] Baguelin, M., S. Flasche, A. Camacho, N. Demiris, E. Miller, and W. J. Edmunds. ‘Assessing Optimal Target Populations for Influenza Vaccination Programmes: An Evidence Synthesis and Modelling Study’. PLoS Med 10, no. 10 (2013): e1001527. https://doi.org/10/gbfntv.

[2] Birrell, Paul J., Richard G. Pebody, André Charlett, Xu-Sheng Zhang, and Daniela De Angelis. ‘Real-Time Modelling of a Pandemic Influenza Outbreak’. Health Technology Assessment (Winchester, England) 21, no. 58 (2017): 1–118. https://doi.org/10.3310/hta21580.

[3] Hao, Xingjie, Shanshan Cheng, Degang Wu, Tangchun Wu, Xihong Lin, and Chaolong Wang. ‘Reconstruction of the Full Transmission Dynamics of COVID-19 in Wuhan’. Nature 584, no. 7821 (August 2020): 420–24. https://doi.org/10.1038/s41586-020-2554-8.

[4] Hill, Edward M., Stavros Petrou, Simon de Lusignan, Ivelina Yonova, and Matt J. Keeling. ‘Seasonal Influenza: Modelling Approaches to Capture Immunity Propagation’. PLOS Computational Biology 15, no. 10 (28 October 2019): e1007096. https://doi.org/10/ghfqrm.

Reviewers' comments:

Reviewer's Responses to Questions

**Comments to the Author**

1. If the authors have adequately addressed your comments raised in a previous round of review and you feel that this manuscript is now acceptable for publication, you may indicate that here to bypass the “Comments to the Author” section, enter your conflict of interest statement in the “Confidential to Editor” section, and submit your "Accept" recommendation.

Reviewer #1: All comments have been addressed

2. Is the manuscript technically sound, and do the data support the conclusions?

Reviewer #1: Partly

3. Has the statistical analysis been performed appropriately and rigorously? 

Reviewer #1: Yes

4. Have the authors made all data underlying the findings in their manuscript fully available?

Reviewer #1: Yes

5. Is the manuscript presented in an intelligible fashion and written in standard English?

Reviewer #1: Yes

6. Review Comments to the Author

Reviewer #1: This manuscript introduces a new observation model that accounts for underreporting of SARS-CoV2 cases by combining an SIR model with a stochastic observation model. In general the manuscript is clear and well written and the methods used are clearly described. I have two main comments though, the first one a technical concern, while the second one has to do with framing of the research.

Firstly the proposed model seems to allow for double counting of infections to occur, because tested individuals are not taken out of the number of real cases (Xn). This is especially a concern on days that the number of counted cases is (almost) as high as the number of actual cases, because on those days none of the cases should carry over to the next day. In that light alpha also seems extremely high, with in many locations, more than 90% of the cases being carried over from one day to the next, which could easily lead to cases being counted many times. Related to this the underreporting rate (1-q) also seems low (below 0.5 in many cases). Especially compared to other estimates of the ascertainment rate (e.g. 0.23 [3]). A low underreporting rate with a high alpha would lead to cases being double counted extremely often.

With regard to the framing of the research. In essence, the work presented here is seems to be about fitting a (simplified) SIR model to the outbreak, using an observation model. This is already an extremely rich field (e.g. 1-4) and many of those models do use observation models of various complexity. As far as I know the presented observation model is new in that it carries over cases from previous days using a stochastic process and the observation probability changes with the day of the week (q). Still I believe the work would be strengthened by a comparison with such models.

Minor comments:

- The logistic function should be represented with logit^-1 not logit (equation 10 and throughout the text)

- In the first supplementary material, equation labels are inconsistent. The equations are labelled using S1.x, while they are referred to (in the text) as A.x.

References:

[1] Baguelin, M., S. Flasche, A. Camacho, N. Demiris, E. Miller, and W. J. Edmunds. ‘Assessing Optimal Target Populations for Influenza Vaccination Programmes: An Evidence Synthesis and Modelling Study’. PLoS Med 10, no. 10 (2013): e1001527. https://doi.org/10/gbfntv.

[2] Birrell, Paul J., Richard G. Pebody, André Charlett, Xu-Sheng Zhang, and Daniela De Angelis. ‘Real-Time Modelling of a Pandemic Influenza Outbreak’. Health Technology Assessment (Winchester, England) 21, no. 58 (2017): 1–118. https://doi.org/10.3310/hta21580.

[3] Hao, Xingjie, Shanshan Cheng, Degang Wu, Tangchun Wu, Xihong Lin, and Chaolong Wang. ‘Reconstruction of the Full Transmission Dynamics of COVID-19 in Wuhan’. Nature 584, no. 7821 (August 2020): 420–24. https://doi.org/10.1038/s41586-020-2554-8.

[4] Hill, Edward M., Stavros Petrou, Simon de Lusignan, Ivelina Yonova, and Matt J. Keeling. ‘Seasonal Influenza: Modelling Approaches to Capture Immunity Propagation’. PLOS Computational Biology 15, no. 10 (28 October 2019): e1007096. https://doi.org/10/ghfqrm.

7. PLOS authors have the option to publish the peer review history of their article (what does this mean?). If published, this will include your full peer review and any attached files.

Reviewer #1: No

---

## [Author Response · Author response to Decision Letter 1]

10 Nov 2020

We thank the referee for careful review which helped to improve the manuscript. Below you will find our replies to the reviewer comments.

<<This manuscript introduces a new observation model that accounts for underreporting of SARS-CoV2 cases by combining an SIR model with a stochastic observation model. In general the manuscript is clear and well written and the methods used are clearly described. I have two main comments though, the first one a technical concern, while the second one has to do with framing of the research.

Firstly the proposed model seems to allow for double counting of infections to occur, because tested individuals are not taken out of the number of real cases (Xn). This is especially a concern on days that the number of counted cases is (almost) as high as the number of actual cases, because on those days none of the cases should carry over to the next day. In that light alpha also seems extremely high, with in many locations, more than 90% of the cases being carried over from one day to the next, which could easily lead to cases being counted many times. Related to this the underreporting rate (1-q) also seems low (below 0.5 in many cases). Especially compared to other estimates of the ascertainment rate (e.g. 0.23 [3]). A low underreporting rate with a high alpha would lead to cases being double counted extremely often.>>

Firstly, we think it is necessary to re-write the paragraph in the paper related to the interpretation of INAR(1) models, as we have realized that it can potentially lead to misunderstanding. In particular, the following modification has been added to the new manuscript (red-highlighted lines - page 3/18):

“The standard interpretation of an INAR(1) model is that an α proportion of the individuals at time t “survive” and are part of the population at time 

t+1. However, this interpretation is misleading in our context. The observations at time t+1 are all new individuals; some correspond to the binomial thinning and the others to the independent innovations. It is known that for many applications for where INAR(1) models can be applied, this meaningful interpretation is not possible. However, the thinning is needed for modelling the autocorrelation of time series. For instance, this is the situation for the example of meningococcal infection analysed in Cardinal et al. (1999).”

With this clarification, we think now is more evident that the INAR(1)-latent process is not interpretable in our application, but it allows modelling a specific auto-correlation of the data. This auto-correlation structure is needed here as the COVID data are intensely temporally correlated, so that this should be considered in the model to mimic the real phenomenon more appropriately. In this sense, the α parameter of the INAR(1)-latent process has to do with this auto-correlation, e.g., high values of this parameter mean that there is a strong relationship between the near past (observation at time t-1) and today (observation at time t). 

We would also like to point out the difference between the latent and observed processes and the parameters that characterize them. High values of the estimated α given in Tables 1 and 2 tell us that the daily number of unobserved COVID cases is strongly auto-correlated, as we expected. Low values of the non-underreporting rate (1-ω) (or high values of the under-reporting rate ω) in Tables 1 and 2 tell us that most of the days, the number of observed COVID cases was under-reported and that we only observed a part of what occurred. In other words, our data are frequently under-reported. If data are frequently under-reported, what we observe most of the time is a part of the INAR(1)-latent process Thus the observed process is likely “away” to the latent process (e.g., the observed process does not necessarily have to be an INAR(1) process). However, and based on the comment regarding the INAR(1) interpretation here, it does not happen that when α and (1-ω) are respectively and simultaneously high and low, we tend to count the same observations more than once. It should also be taken into account that parameters α and (1-ω) are related to different processes; α is related to the latent process and (1-ω) to the observed process. 

<<With regard to the framing of the research. In essence, the work presented here is seems to be about fitting a (simplified) SIR model to the outbreak, using an observation model. This is already an extremely rich field (e.g. 1-4) and many of those models do use observation models of various complexity. As far as I know the presented observation model is new in that it carries over cases from previous days using a stochastic process and the observation probability changes with the day of the week (q). >>

The model presented here is not a simplified version of a SIR model but a combination of integer-valued time series and SIR models to estimate under-reporting and epidemic evolution simultaneously. 

In particular, our model consists of two processes X_n and Y_n; the first one is the unobserved process which follows an INAR(1) structure, while the second one is the observed process that is equal to the INAR(1) latent process with probability 1- ω, or a thinning operator of the latent-INAR(1) process with probability ω (see equation X). Therefore, the proposed model is based on an integer-valued time series approach.

However, as the INAR(1) model usually assumes stationarity (in particular, constant mean), and this is obviously not the case, hence we incorporate a time-varying mean to the innovations of the INAR(1) model defined as an approximation to the solution of the SIR model, that we have also presented in the paper. 

In addition, our model's primary goal consists of estimating the daily non-ascertained cases in this time-varying setting from the official daily confirmed cases. Observe that to estimate the global number of non-ascertained cases through a compartmental SIR (or SEIR) model, data at an individual level are needed (e.g., for each individual, we need to have information on the date of the first symptom, the number of days in quarantine, in hospital, intensive care, recovery, etc.). In our case, however, we estimate the daily number of non-ascertained cases only using the daily number of confirmed cases.

<<Still I believe the work would be strengthened by a comparison with such models.>>

Articles provided by the referee have been carefully read and compared to our model. Although some of these works do not handle under-reporting in data (as it does our proposed model), they offer interesting applications through SEIR models that can inspire better model COVID data's evolution and policy makers. 

In particular, Baguelin et al. (2013) use an SEIR model to evaluate vaccine policies' effects on England and Wales's influenza epidemics. This work's aim is not on the estimation of non-reported influenza cases in England and Wales, but it may serve as an example of how to evaluate the effect of vaccination on the COVID epidemic. Similarly, Hill et al. (2019) also employ an SEIR model to study seasonal influenza evolution in England. The particularity of that work is that the authors link the prior seasonal information to the immunity in the following period to ensure non-independence between the successive influenza seasons. In the same lines, the research report by Birrell et al. (2017) is a set of different research works aimed at modelling the influenza epidemics in England, but most of them are not focused on the misreporting estimation. 

As the three works above add different perspectives for pandemics' evolution modelling, the following lines have been included in the manuscript:

“The key interest of researchers when dealing with an epidemic such as the current SARS-CoV2 is to estimate the propagation of the disease and predict its possible end date to apply appropriate measures of control and prevention Bedford et al. (2020). The literature offers different approaches to deal with so, as the so-called SIR and SEIR compartmental models. These models have extensively used for study influenza's epidemic evolution as Baguelin et al. (2013) who use an SEIR model to evaluate vaccine policies effects on England and Wales's influenza epidemics, Hill et al. (2019) who employs SEIR model to study seasonal influenza evolution in England by linking the prior seasonal information to the immunity in the following period in order to ensure non-independence between the successive influenza seasons, or Birrell et al. (2017) who presents a set of different research works aimed at modelling the influenza epidemics.”

Finally, the paper by Hao et al. (2020) is the closest to our model among the four proposed. This paper presents an SEIR model that considers a compartment for non-ascertained individuals. In doing so, the authors can estimate those COVID infections that have not been officially registered. However, there are essential particularities between the model by Hao et al. (2020) and our model that have to be highlighted. First, the model by Hao et al. (2020) needs data at an individual level, but our model perfectly works with aggregated data on the confirmed COVID infections. Also, the SEIR model needs information related to infections, type of symptoms, days of quarantines, hospitalizations, etc. but our model only needs the number of daily confirmed cases. Second, our model introduces a particular type of auto-correlation structure of the data that the SEIR model by Hao et al. (2020) does not. In other words, our model takes completely different assumptions compared to the model by Hao et al. (2020), and thus both models can be seen as alternatives to one of the other. Last, our model allows estimating the number of unobserved cases at a quasi-real-time (at a daily basis), while the model by Hao et al. (2020) gives an aggregated estimation of the total amount of infections that were unobserved. 

In order to point this out in our paper, the following information has been included:

“Global percentages of under-reporting during a given period of time can be estimated, for instance, with stochastic Susceptible-Exposed-Infectious-Recovered (SEIR) models, including unobservable compartments of non-ascertained individuals. In order to estimate the parameters in such models, it is necessary to have data on individual evolution of the epidemic, that is, for each individual, the date of contagion or appearance of first symptoms, number of days in quarantine, hospital, or similar information is required, regardless of the estimation methodology. There are many examples of this situation. For instance, Hao et al. (2020) who estimates SARS-CoV2 in Wuhan via MCMC, Cabaña (2020) who uses least squares estimator for SARS-CoV2 in Uruguay, or Ducrot et al. (2020) who employs a simpler version of an SEIR model, called Susceptible-Infected-Recovered (SIR) model, to understand the relationship between the observed and unobserved cases of the Hong Kong seasonal influenza epidemic in New York between 1968 and 1969.

Although the previous works proposed new methods to describe, identify or estimate under-reporting of data, none of them, to our knowledge, tried to model the under-reporting in integer-valued time series data.”

Minor comments:

<<The logistic function should be represented with logit^-1 not logit (equation 10 and throughout the text)>>

We have removed the mention to logit in the text to avoid confusion. 

<<In the first supplementary material, equation labels are inconsistent. The equations are labelled using S1.x, while they are referred to (in the text) as A.x.>>

References in the text are of the form S.x. now.

---

## [Editor Report · Decision Letter 2]

13 Nov 2020

Estimating the real burden of disease under a pandemic situation: The SARS-CoV2 case

PONE-D-20-20748R2

Dear Dr. Fernández-Fontelo,

We’re pleased to inform you that your manuscript has been judged scientifically suitable for publication and will be formally accepted for publication once it meets all outstanding technical requirements.

Kind regards,

Paul K. Newton, Ph.D.

Academic Editor

PLOS ONE
---

## [Editor Report · Acceptance letter]

20 Nov 2020

PONE-D-20-20748R2 

Estimating the real burden of disease under a pandemic situation: The SARS-CoV2 case 

Dear Dr. Fernández-Fontelo:

I'm pleased to inform you that your manuscript has been deemed suitable for publication in PLOS ONE. Congratulations! Your manuscript is now with our production department. 

Kind regards, 

on behalf of

Professor Paul K. Newton 

Academic Editor

PLOS ONE